# HSF-IBI: A Universal Framework for Extracting Inter-Beat Interval from Heterogeneous Unobtrusive Sensors

**DOI:** 10.3390/bioengineering11121219

**Published:** 2024-12-02

**Authors:** Zhongrui Bai, Pang Wu, Fanglin Geng, Hao Zhang, Xianxiang Chen, Lidong Du, Peng Wang, Xiaoran Li, Zhen Fang, Yirong Wu

**Affiliations:** 1School of Electronic Information and Electrical Engineering, Shanghai Jiao Tong University, Shanghai 200240, China; zhongrui.bai@sjtu.edu.cn; 2Aerospace Information Research Institute, Chinese Academy of Sciences, Beijing 100190, China; wupang17@mails.ucas.ac.cn (P.W.); gengfanglin20@mails.ucas.ac.cn (F.G.); zhanghao190@mails.ucas.ac.cn (H.Z.); chenxx@aircas.ac.cn (X.C.); lddu@mail.ie.ac.cn (L.D.); wangpeng01@aircas.ac.cn (P.W.); wyr@mail.ie.ac.cn (Y.W.); 3Beijing Friendship Hospital, Capital Medical University, Beijing 100050, China; 4Personalized Management of Chronic Respiratory Disease, Chinese Academy of Medical Sciences, Beijing 100190, China

**Keywords:** inter-beat interval, ballistocardiography, unobtrusive cardiac vibration signal, template matching

## Abstract

Heartbeat inter-beat interval (IBI) extraction is a crucial technology for unobtrusive vital sign monitoring, yet its precision and robustness remain challenging. A promising approach is fusing heartbeat signals from different types of unobtrusive sensors. This paper introduces HSF-IBI, a novel and universal framework for unobtrusive IBI extraction using heterogeneous sensor fusion. Specifically, harmonic summation (HarSum) is employed for calculating the average heart rate, which in turn guides the selection of the optimal band selection (OBS), the basic sequential algorithmic scheme (BSAS)-based template group extraction, and the template matching (TM) procedure. The optimal IBIs are determined by evaluating the signal quality index (SQI) for each heartbeat. The algorithm is morphology-independent and can be adapted to different sensors. The proposed algorithm framework is evaluated on a self-collected dataset including 19 healthy participants and an open-source dataset including 34 healthy participants, both containing heterogeneous sensors. The experimental results demonstrate that (1) the proposed framework successfully integrates data from heterogeneous sensors, leading to detection rate enhancements of 6.25 % and 5.21 % on two datasets, and (2) the proposed framework achieves superior accuracy over existing IBI extraction methods, with mean absolute errors (MAEs) of 5.25 ms and 4.56 ms on two datasets.

## 1. Introduction

In the past decade, unobtrusive vital sign monitoring (where users are unaware of being monitored and do not need to wear or attach devices) has gained widespread attention from both the academic and industrial communities as an emerging healthcare technology. Unlike traditional biomedical sensors, such as electrocardiogram (ECG) and photoplethysmography (PPG), which require mechanical contact with the skin, contactless and wireless sensors can measure physiological signals by detecting body vibration or electromagnetic waves reflected by the body [1,2]. The new unobtrusive approaches eliminate the need for electrode attachments and long electrode wires. These approaches have many potential medical and healthcare applications, such as improving patient compliance, reducing the risk of exposure to infection, and assessing the severity and prognosis of diseases [2,3].

Utilizing these unobtrusive technologies allows for the measurement of inter-beat intervals (IBIs), which, in turn, allows for the calculation of heart rate variability (HRV). Compared to average heart rate (HR), HRV provides more detailed insights into cardiac and neurological functions. With respect to the types of unobtrusive sensors used for measuring IBIs, the technology spectrum includes bed-embedded piezoelectric, fiber optic sensors or force sensors for measuring ballistocardiogram (BCG) [4,5,6,7], biometric radar sensors for Doppler cardiography (DCG) [8,9,10,11,12,13], and a camera for remote photoplethysmography (rPPG) and remote ballistocardiogram (rBCG) [14]. Among these options, DCG and bed-based BCG are not affected by lighting conditions and privacy concerns, making them ideal for long-term monitoring during the night.

BCG and DCG are both physiological signals that reflect cardiac mechanical activity. While BCG mainly reflects the impact force of the aorta and DCG reflects chest wall displacement, both signals are complex combinations of circulatory responses and myocardial vibration components [12,15]. Therefore, they can be collectively referred to as *cardiac vibration signals* (CVSs) in a broader sense [16]. Additional modalities that could fall into the category of CVS include video-based rBCG, as well as those employing Wi-Fi [17], radio-frequency identification (RFID) [18], and smart speaker [19] technologies for cardiac motion acquisition.

However, despite their unobtrusive advantages, CVSs are more prone to motion artifacts compared to contact-based monitoring methods like ECG and PPG. Furthermore, the placement of sensors in proximity to the body or heart plays a crucial role in determining the signal-to-noise ratio. This leads to reduced accuracy and coverage in detecting IBIs, which is unacceptable for reliable clinical applications based on HRV, such as cardiovascular event risk analysis and sleep stage classification [20]. Overall, due to the limited reliability of unobtrusive monitoring methods in detecting IBIs, their application in various monitoring scenarios has been greatly restricted.

Multi-channel sensors have been shown to enhance accuracy and coverage in unobtrusive heart monitoring compared to single-channel CVS detection [16,21], primarily due to the spatial distribution of different sensors. Building on this concept, our work further explores the idea of simultaneously leveraging the spatial distribution and the unique characteristics of various sensor types, such as their varying sensitivities to distinct types of noise. For example, in BCG measurement, piezoelectric sensors are sensitive to electromagnetic noise in the environment, surges in the circuit, and other electrical interference [22]. Load cells placed on bed legs are more sensitive to ground vibrations (such as footsteps). In contrast, noise and object movement in the radar coverage area affect the DCG signal [1]. Therefore, it can be inferred that the performance of IBI detection may be further improved by utilizing the different characteristics of different types of sensors. Because of the improvement in terminal hardware performance, it is feasible to process multi-channel signals from various contactless sensors. Furthermore, given the rapid development of unobtrusive vital sign monitoring, it is appealing to perform more robust and accurate heart rate monitoring using heterogeneous sensors. At the same time, different sensors may introduce more diverse signal characteristics and morphologies, posing a challenge to signal processing methods.

In this study, we proposed a novel framework—HSF-IBI (Heterogeneous Sensor Fusion-Inter-Beat Interval)—for accurately extracting IBIs from multichannel CVSs, and we applied it to heterogeneous sensors’ data. The main contributions of this paper can be summarized as follows:This paper is the first to perform higher performance IBI measurement on heterogeneous unobtrusive sensor-collected CVSs.A novel multi-channel algorithm framework for extracting IBIs from CVSs is designed. This framework is based on principles of template matching and, for the first time, introduces the harmonic summation (HarSum) algorithm, basic sequential algorithmic scheme (BSAS), optimal band selection (OBS), and signal quality index (SQI)-based beat-to-beat signal fusion scheme to obtain more robust and accurate IBI estimation.The proposed framework was evaluated on two datasets, including a self-collected dataset and an open-source dataset. Experimental results demonstrate that the proposed method can adapt to various signal types and morphologies. Additionally, the results show that the IBI detection rate and accuracy on heterogeneous sensor data outperform those of single-channel and single-modality sensor data, and the proposed framework achieves state-of-the-art accuracy for unobtrusive IBI estimation.

The rest of the paper is organized as follows. In Section 2, relevant work on CVS-based IBI measurement is discussed. Section 3 describes the proposed HSF-IBI algorithm framework. Next, Section 4 outlines the datasets used in this study, the evaluation metrics, and the evaluation scenarios. Section 5 presents the experimental results and discusses the findings. Finally, the paper is concluded in Section 6.

## 2. Related Works

Unobtrusive vital sign monitoring has led to the development of various algorithms for IBI estimation from CVSs, with most focusing on single-channel sensing signals. Unlike ECG signals, heartbeats in CVSs lack prominent features like the R-wave and vary significantly depending on sensor type, placement, subject, and posture [1]. These challenges have motivated the development of more advanced algorithms.

### 2.1. Single-Channel CVS-Based IBI Estimation

Traditional methods for IBI estimation often rely on preprocessing techniques, such as empirical mode decomposition (EMD) and wavelet transform (WT), to detect local maxima as heartbeat locations [3,11]. While straightforward, these methods struggle with the variability of CVS signals, as preprocessing techniques optimized for specific sensors may fail on others. Furthermore, smoothing during preprocessing can reduce heartbeat detection accuracy.

Template matching (TM), a widely used pattern recognition technique, determines the similarity between a template and a target signal [23]. In IBI estimation, TM extracts a heartbeat template and computes its similarity with the remaining signal to locate heartbeats [3,23,24]. However, recent TM-based methods [24] often require ECG-labeled training for BCG signals, and the morphological variability of signals across subjects necessitates individual training for each subject.

The continuous local interval estimation (CLIE) algorithm [4] represents a probabilistic approach, using Bayesian fusion to combine results from short-time estimators, with the probability density function (PDF) peak serving as the candidate heartbeat interval. While CLIE avoids the need for explicit templates, its accuracy is limited by its assumption of consistent heartbeat morphology across adjacent intervals, as this assumption does not always hold. Additionally, its probabilistic nature can result in less precise IBI estimates.

Waveform feature-based methods focus on identifying specific points or shapes within the signal. For example, Sakamoto et al. [25] identified six key features from UWB radar signals and used topological characteristics to extract IBIs, while Bruser et al. [26] employed machine learning to classify heartbeat shapes for IBI estimation. However, these methods are highly dependent on predefined waveform features, limiting their flexibility and generalizability.

More recently, deep learning models such as Bi-LSTM and Conv-Transformer have been applied to IBI estimation [27,28]. These methods show promise, especially in processing low-quality signals, but they have not yet achieved highly competitive performance in this task. In fact, the significant morphological variability of CVSs across datasets and subjects challenges deep learning methods, limiting their ability to adapt and generalize effectively.

### 2.2. Multi-Channel CVS-Based IBI Estimation

Multi-channel methods aggregate information from multiple signals to improve IBI estimation. For example, some studies use the maximum amplitude channel or an average of all channels to compute IBIs [24,29]. While these approaches outperform single-channel methods, they do not fully leverage the potential of multi-channel data.

Bruser extended Bayesian fusion to multi-channel BCGs [16], evaluating it on 28 night-long recordings collected with an eight-channel PVDF sensor array. The proposed xCLIE method fused PDFs from the eight channels, reducing IBI error from 2.2% to 1.0% and increasing coverage (the percentage of the recording period analyzed) from 68.7% to 81.0%. Another study [21] refined the xCLIE approach by introducing Gaussian weighting and modified q-values to create a weighted joint PDF. Evaluated on data from 25 individuals using a four-channel load cell bed, this method achieved 98.69% coverage and a 2.74% average relative error during rest and supine conditions.

Despite these advances, multi-channel CLIE-based methods remain limited by the assumptions and constraints of the original CLIE algorithm, including its reliance on probabilistic estimations and morphological consistency.

In this paper, we propose a novel framework, HSF-IBI, to overcome the limitations of existing methods for estimating IBIs from multi-channel CVSs.

## 3. Methods

The core concept of the proposed HSF-IBI algorithm framework is to compute and select IBIs of heartbeats based on the SQI of each signal segment and individual heartbeat. The overall process is illustrated in Figure 1.

### 3.1. Preprocessing and Optimal Band Selection

Our prior work has shown that different methods for preprocessing BCGs can lead to varying IBI estimation outcomes [30]. For certain BCGs, higher filter frequencies can yield more accurate IBI estimations when ECG-based IBIs are used as the ground truth. This can be explained by heart valve signal (HVS) theory, where high-frequency components originate from heart valve vibrations rather than aortic or torso movement [31]. A similar phenomenon is observed in radar-measured DCGs, where appropriate frequency filtering produces signals similar to phonocardiograms [32]. Signals closer to the heart tend to improve IBI estimation accuracy by providing sharper peaks and reducing timing errors [33].

However, not all heart vibration signals contain HVS. BCGs closer to the heart are more likely to include high-quality HVS, while those further away may only contain low-frequency signals from aortic or torso movement. Additionally, due to variations in subjects and sensors, heart valve signals may fall within different frequency bands. Thus, it is essential to assess the suitability of signals for HVS extraction and determine the feasible frequency range.

To achieve OBS, we employ a bandpass filter BPF_OBS_ with an adjustable frequency range. To preserve phase invariance, the filtering is applied bidirectionally. Following [31], the high-frequency cutoff is fixed at $f_{H}=40$ Hz.

To reduce computational complexity, four low-frequency cutoff frequencies are heuristically selected:  
(1)$$f_{L}\in F=\left\{f_{1},f_{2},f_{3},\ldots \right\}$$

Considering that components at higher frequencies encompass a greater proportion of the cardiac valve signatures, leading to more accurate IBI outcomes, the signal processed by the filter with a higher $f_{L}$ is assigned a higher weight β (to be used in Section 3.5) according to
(2)$$\beta =1+K\times \left(f_{L}-1\right),$$
where *K* is a weighting coefficient.

### 3.2. Mean HR Estimation Based on HarSum-HR

There are two main approaches to calculating mean heart rate from physiological signals—time-domain and frequency-domain methods. Time-domain methods detect heartbeat feature points (e.g., R waves in ECG) within a specific time frame to compute the average heart rate based on IBIs. However, the lack of distinct feature points in CVSs makes this approach difficult and error-prone. Frequency-domain methods preprocess the signal to compute its envelope, perform a time–frequency transformation to obtain a spectrogram, and identify the heart rate in the spectrum. Yet, the variability in CVS morphology challenges fixed-parameter preprocessing methods, complicating envelope extraction.

To address these issues, we propose the HarSum-HR method, which utilizes the harmonic summation algorithm adapted from fundamental frequency estimation in speech [34,35]. The HarSum-HR method involves the following steps: normalization and preprocessing of the raw signal, calculation of its Fast Fourier Transform (FFT) spectrum, computation of the HarSum spectrum, and estimation of the mean heart rate from the HarSum spectrum.

Step 1: Normalization and preprocessing. All signals are processed with an 8 s window and a 1 s step size. After z-score normalization, the signals are squared and filtered using a band-pass filter (BPF_HarSum_), which shifts frequencies to the range of lower-order harmonics. Step 1 can be represented as:(3)$$y\left(t\right)={\mathrm{BPF}}_{\mathrm{HarSum}}{\left[\mathrm{Norm}\left(x\left(t\right)\right)\right]}^{2}.$$

Step 2: time–frequency transformation. In this step, we apply a Hamming window to each segment of the signal, and then compute its FFT spectrum:(4)$$X\left(f\right)=\mathrm{FFT}\left[y\left(t\right)*\mathrm{Hamming}(8s)\right].$$

Step 3: Compute the HarSum spectrum. As shown in Figure 2a, the obtained FFT spectrum contains several harmonics, making it difficult to identify the heart rate’s fundamental frequency accurately. However, it can be observed that in the spectrum, a good candidate fundamental frequency *f* should not only have peaks at integer multiples of f but also valleys at half-integer multiples of *f*. The concept of peak–valley distance (PVD) is introduced to measure the significance of the peak at frequency *f* relative to its two adjacent valleys in its amplitude spectrum [35]:(5)$$d_{k}\left(f\right)=\left|X\left(kf\right)\right|-\frac{1}{2}\left|X\left(\left(k-1/2\right)f\right)\right|-\frac{1}{2}\left|X\left(\left(k+1/2\right)f\right)\right|$$

Here, $d_{k}\left(f\right)$ represents the PVD at the *k*-th harmonic of *f*. $\left|X\left(kf\right)\right|$ represents the amplitude of the frequency response at a frequency of kf, and *k* is a positive integer. The average peak-to-valley distance (APVD) of each harmonic can represent the significance of frequency *f*:  
(6)$$D_{k}\left(f\right)=\frac{1}{n}\sum_{k=1}^{n}d_{k}\left(f\right).$$

The APVD is expressed as the inner product of the amplitude spectrum and a kernel function, which has upward impulses at integer multiples of *f* and downward impulses at half-integer multiples of *f*, as shown in Figure 2a. The number of harmonics is limited to seven or fewer. For a frequency point $f^{\prime }$, the kernel function is defined as:(7)$$K\left(f;f^{\prime }\right)=\sum_{i\in P}K_{i}\left(f;f^{\prime }\right),$$
where P={1,2,3,5,7} is the set of prime numbers no greater than seven, and 
(8)$$K_{i}\left(f;f^{\prime }\right)=\left\{\begin{array}{cc} \frac{1}{\sqrt{i}} & \mathrm{if}\;f=i\times f^{\prime }\\ -\frac{1}{2\sqrt{i}} & \mathrm{if}\;f=\left(i\pm 1/2\right)\times f^{\prime }\\ 0 & \mathrm{otherwise}. \end{array}\right.$$

Following SWIPE’ [36], a decreasing weight is applied to harmonics to prevent higher harmonics from achieving the same score as the fundamental frequency. Prime numbers are used to reduce misclassification of the fundamental frequency as a harmonic.

For each candidate frequency $f^{\prime }$, the HarSum spectrum is computed as the inner product of the FFT amplitude spectrum and the kernel function:(9)$$|HS\left(f^{\prime }\right)|=<X\left(f\right),K\left(f;f^{\prime }\right)>.$$

Step 4: Mean heart rate from the HarSum spectrum. As shown in Figure 2b, the highest value in the HarSum spectrum represents the average heart rate. Figure 2c,d demonstrate that HarSum provides a clearer representation of heart rate in the spectrogram.

### 3.3. ${SQI}_{HarSum}$ and Channel Selection

Here, we can obtain a signal quality index ${SQI}_{HarSum}$ based on the HarSum spectrum. Since the signal has been normalized, the amplitude of the HarSum spectrum at candidate fundamental frequency points can, to some extent, represent the energy ratio of the candidate fundamental frequency and its corresponding higher harmonics. We define:(10)$${SQI}_{HarSum}=\max\left(HarSum\left(f\right)\right).$$

For each channel’s CVS signal *s*, only one $f_{L}$ frequency in Equation (Equation 1) is selected for the subsequent template extraction. This selection follows the scheme below:

If there exists a signal with ${\mathrm{SQI}}_{\mathrm{Harsum}}>Q_{1}$, choose the highest $f_{L}$ that satisfies the condition of $SQI_{\mathrm{HarSum}}\left(f_{L}\right)$ greater than $Q_{1}$:$$max\{f_{L}|SQI_{\mathrm{HarSum}}\left(s,f_{L}\right)>Q_{1}\}$$

If no signal quality exceeds $Q_{1}$ but at least one is greater than $Q_{2}$, select the $f_{L}$ with the highest quality:$$\arg\max_{f_{L}\in \left\{f_{L}\right\}}\left\{SQI_{\mathrm{HarSum}}\left(s,f_{L}\right)|Q_{1}<SQI_{\mathrm{HarSum}}\left(s,f_{L}\right)<Q_{2}\right\}.$$

If none are greater than $Q_{2}$, discard this channel.

Here, $Q_{1}$ and $Q_{2}$ are two threshold values for $SQI_{\mathrm{HarSum}}$. Through such channel and frequency band screening, the optimal frequency band can be selected while ensuring the signal quality.

### 3.4. Template Group Extraction Based on BSAS

For each channel meeting the $SQI_{HarSum}$ criteria, templates are extracted to address respiratory modulation effects. Instead of a single template, a group $\Psi ={\left\{T_{i}\right\}}_{i=1}^{2}$ is used.

All local maxima ${pindex_{i}}_{i=1}^{N}$ within the first 8 s of signal S are identified, and a segment $q_{i}$ of length *L*, centered at $pindex_{i}$, is extracted. *L* is calculated as Equation (Equation 11), where SR is the sampling rate of the CVSs.
(11)$$L=\frac{60}{HR_{mean}}\times SR.$$

Then, the correlation matrix for ${\left\{q_{i}\right\}}_{i=1}^{N}$ is calculated, resulting in the following matrix:(12)$$\mathrm{CorrMat}={\left[\mathrm{corr}(q_{i},q_{j})\right]}_{i,j=1}^{N},$$
where $\mathrm{corr}(q_{i},q_{j})$ denotes the Pearson correlation coefficient between $q_{i}$ and $q_{j}$.

The correlation coefficients are clustered using the basic sequential algorithmic scheme (BSAS) with a threshold θ, grouping signal segments with similar morphology into *m* clusters. The cluster $C_{s}$ with the highest average center point value is then selected:(13)$$C_{s}=\arg\max_{C_{j}\subseteq \left\{C_{1},C_{2},\ldots ,C_{m}\right\}}\left\{\frac{\sum_{q_{i}\in C_{j}}q_{i}\left(\frac{L}{2}\right)}{N_{j}}\right\},$$
where $N_{j}$ denotes the number of samples in cluster $C_{j}$.

Then, choose a template $q_{s}$ from it based on:(14)$$q_{s}=\arg\max_{q_{j}\in C_{s}}\left\{\sum_{\begin{array}{c} q_{i}\in C_{s},q_{i}\neq q_{j} \end{array}}\mathrm{corr}\left(q_{j},q_{i}\right)\right\}.$$

Equation (Equation 14) selects the q with the highest correlation sum within its cluster, representing the most similar signal. Unlike [37], we avoid averaging signals within a cluster to preserve morphology.

The first template is obtained as $T_{1}=q_{s}$. A second template is calculated if the following condition is met:(15)$$\exists h\in N,0<h<h_{s},pindex_{h_{s}}^{C_{s}}-pindex_{h_{s-1}}^{C_{s}}>1.5L.$$

Here, $pindex_{h_{s}}^{C_{s}}$ denotes the midpoint (or peak) position of sample $h_{s}$ in the first selected cluster $C_{s}$.

Equation (Equation 15) identifies a significant gap between two consecutive samples in $C_{s}$, indicating the presence of other heartbeats in *S* with shapes distinct from those in $C_{s}$. If condition (Equation 15) is satisfied, an empty set Γ is initialized to record the positions of missing signal segments in S.
(16)$$\Gamma =\Gamma \cup \{pindex_{h}|pindex_{h_{s-1}}^{C_{s}}<pindex_{h}<pindex_{h_{s}}^{C_{s}}\}.$$

The clusters $\left\{C_{j_{s}}\right\}$ containing the pindex located in the missing segments of the $C_{s}$ signal are selected, excluding the $C_{s}$ cluster, according to the following equation:(17)$$\exists h^{\prime }\in N,0<h^{\prime }<h_{j},pindex_{h^{\prime }}^{C_{j}}\in \Gamma ,\left(j^{\prime }\neq s\right).$$

Equations (Equation 13) and (Equation 14) are then applied to select a second cluster $C_{s}$ and template $T_{2}$ from $\left\{C_{j_{s}}\right\}$.

Although additional templates can be extracted, practice shows that two templates per channel are typically sufficient, as respiration generally involves only two phases: inhalation and exhalation. Adjusting the threshold θ can modify cluster similarity.

The process of extracting a template group from a single channel’s signal, as described in Section 3.4, is summarized in Algorithm 1.
**Algorithm 1** Template group calculate for one channel  **Input:** S: the initial 8-second CVS, $HR_{mean}:$ mean HR, SR: sample rate  **Output:** $\Psi ={\left\{T_{j}\right\}}_{j=1}^{2}:$ template group1:$L\leftarrow 60/HR_{mean}$2:**for** each maximum $pindex_{i}$ in S **do**3:      $q_{i}\leftarrow S\left[\left(pindex_{i}-L/2\right):\left(pindex_{i}+L/2\right)\right]$4:**end for**5:Calculate the correlation coefficient matrix CorrMat of ${\left\{q_{i}\right\}}_{i=1}^{N}$  6:m←1, $C_{m}\leftarrow \left\{q_{0}\right\}$7:**for** each $q_{i}$ **do**▹ BSAS8:      Find $C_{k}$: $M\left(q_{i},C_{k}\right)\leftarrow \max_{1\leq j\leq m}M\left(q_{i},C_{j}\right)$9:      **if** $M(q_{i},C_{k})<\theta$ **then**10:         Create a new cluster: m←m+1, $C_{m}\leftarrow \left\{q_{i}\right\}$11:    **else**12:        $C_{k}\leftarrow C_{k}\cup \left\{q_{i}\right\}$13:    **end if**14:**end for**15:   ▹ Calculate a template from a cluster16:Find $C_{s}$ using Equation (Equation 14)17:Find $q_{s}$ from $C_{s}$ using using Equation (Equation 15)18:$T_{1}\leftarrow q_{s}$  19:   ▹ Calculate another template if necessary20:Γ←{}21:**for** each $pindex_{h-1},pindex_{h}$ in $C_{s}$ **do**22:       **if** $(p_{index_{h}}-p_{index_{h-1}})>1.5L$ **then**23:             $\Gamma \leftarrow \Gamma \cup \{pindex\;|pindex_{h-1}^{C_{s}}<pindex<pindex_{h}^{C_{s}}\}$24:       **end if**25:**end for**26:**if** 
Γ≠Ø 
**then**27:       Find $\left\{C_{j^{\prime }}\right\}:\exists \;h^{\prime }:\left(0<h^{\prime }<h\right),pindex_{h^{\prime }}^{C_{j^{\prime }}}\in G,\left(j^{\prime }\neq s\right)$28:       Find $C_{s^{\prime }}$ and $T_{2}$: repeat lines 15–18 using $\left\{C_{j^{\prime }}\right\}$29:**end if**

$M\left(q_{i},C_{j}\right)=\min_{1\leq a\leq n_{j}}\mathrm{corr}\left(q_{i},q_{a}^{j}\right)$, represents the minimum correlation between signal segment $q_{i}$ and all signal segments in cluster $C_{j}$, which ensures a relatively high correlation among all signal segments within each cluster.

### 3.5. Heartbeat Detection and IBI Estimation

In the heartbeat detection phase, the method calculates the correlation coefficient between each template within the template set and every peak value in the subsequent CSV signal, resulting in correlation coefficient functions (ccf).
(18)$$ccf_{\alpha }\left(x\right)=\mathrm{corr}\left[\mathrm{cvs}\left(x\right),T_{\alpha }\times P\left(x\right)\right],\alpha \in \left\{1,2\right\}.$$

Here, P(x) is an indicator function:(19)$$P\left(x\right)=\left\{\begin{array}{cc} 1, & \mathrm{if}\;x\;\mathrm{is}\;a\;\mathrm{peak}\;\mathrm{position}\\ 0, & \mathrm{otherwise}. \end{array}\right.$$

Subsequently, local peak positions $x_{\mathrm{peak}}$ are identified within the first 2 s of the ccf signal and in the range [(1−v)L,(1+v)L] following each detected heartbeat position. These peaks are used to determine heartbeat locations and compute the IBIs. Here, *v* is a parameter representing short-term HRV, set to 0.3 in this work. The IBI is then calculated as:(20)$${\mathrm{IBI}}_{\alpha }^{i}=\frac{{\mathrm{peak}}_{\alpha }^{i}-{\mathrm{peak}}_{\alpha }^{i-1}}{\mathrm{SR}}.$$

As for choosing between $ccf_{1}$ and $ccf_{2}$, the reliability of consecutive two heartbeats in the ccf must be considered. When $ccf_{\alpha }\left(x={\mathrm{peak}}_{\alpha }^{i-1}\right)>\theta$ is satisfied, the higher $ccf_{\alpha }\left(x={\mathrm{peak}}_{\alpha }^{i}\right)$ value is selected. At this point, another signal quality index, ${\mathrm{SQI}}_{\mathrm{beat}}$ is represented as:(21)$${\mathrm{SQI}}_{\mathrm{beat}}^{i}=\max\left[ccf_{1}\left(x={\mathrm{peak}}_{1}^{i}\right),ccf_{2}\left(x={\mathrm{peak}}_{2}^{i}\right)\right]\times \beta .$$

Here, the coefficient β is a weight factor associated with the filtering parameters of the current signal, as detailed in Section 3.1. If the ${\mathrm{SQI}}_{\mathrm{beat}}$ of a specific channel falls below the threshold for five consecutive beats, or if the ${\mathrm{SQI}}_{\mathrm{Harsum}}$ of all channels fails to meet the threshold, the template group is updated.

This step focuses solely on peak points rather than matching every sample point of the signal, significantly reducing computational load while ensuring that heartbeat positions correspond to the peak points.

### 3.6. Merging IBIs from Multichannel CVSs

In this step, the IBIs from multiple channels are combined based on the selected ${\mathrm{SQI}}_{\mathrm{beat}}$ of each channel, similar to the previous Section. For the *i*-th heartbeat, when ${\mathrm{SQI}}_{\mathrm{beat}}^{i-1,\mathrm{channel}}>\theta$ is satisfied, select the channel for IBI according to the following expression:(22)$${\mathrm{Selectedchannel}}_{i}=\arg\max_{\mathrm{channel}}\left({\mathrm{SQI}}_{\mathrm{beat}}^{i,\mathrm{channel}}\right).$$

The final IBI is:(23)$${\mathrm{IBI}}_{\mathrm{selected}}^{i}={\mathrm{IBI}}_{\mathrm{Selectedchannel}}^{i}.$$

Figure 3 shows an example of combining multiple channels of IBI from Dataset A and Dataset B.

### 3.7. Parameter Determination

Several parameters are specified in this paper:(1)$f_{L}$ of BPF_OBS_ (in Section 3.1): The candidate low-pass cutoff frequency is set to $f_{L}\in F=\left\{1\;\mathrm{Hz},3\;\mathrm{Hz},5\;\mathrm{Hz},7\;\mathrm{Hz}\right\}$. The frequencies are chosen to ensure that 1 Hz retains low-frequency components of the heartbeat, while 7 Hz effectively filters them out.(2)Filter parameters of BPF_HarSum_ (in Section 3.2): The BPF_HarSum_ retains low-order harmonic components of the heartbeat while removing respiratory interference and high-frequency noise. Its low and high cutoff frequencies are set to 1 Hz and 10 Hz, respectively, corresponding approximately to the first to seventh harmonics of heart rate frequency. Both BPF_OBS_ and BPF_HarSum_ are third-order Butterworth filters.(3)Weighting coefficient *K* (in Section 3.1) and thresholds for ${SQI}_{HarSum}$, $Q_{1}$, and $Q_{2}$ (in Section 3.3): Higher values of *K*, $Q_{1}$, and $Q_{2}$ improve accuracy but reduce IBI coverage, and vice versa. In this paper, *K* is set to 0.05, with $Q_{1}$ and $Q_{2}$ set to 0.52 and 0.26, respectively.(4)Threshold for ${SQI}_{beat}$ (θ) (in Section 3.4): This threshold reflects the matching degree between the template and the current heartbeat. Variations in respiratory and cardiac cycles may reduce the matching score. Here, θ is set to 0.75.(5)Search Range Parameter *v* (in Section 3.5): This parameter addresses instantaneous IBI variability between heartbeats. A value of 0.3 is chosen to account for this variability.

These hyperparameters are derived from prior physiological knowledge and remain consistent across all test subjects.

## 4. Evaluation

### 4.1. Data Description

We evaluated the proposed approach on two datasets: a self-collected dataset and an open-source dataset, both comprising data from heterogeneous unobtrusive sensors.

#### 4.1.1. Dataset A

Dataset A includes CVSs measured using radar and a piezoelectric ceramic (PEC) sensor from 19 healthy individuals in a laboratory setting. Each participant performed 5 min of cycling at 25 km/h on a bicycle ergometer, followed by 4 min of supine rest during which DCG, BCG, and ECG data were synchronously collected.

DCGs were acquired using a 77GHz FMCW radar (AWR1642EVM) and its data acquisition board (DCA1000EVM, Texas Instruments, Dallas, TX, USA) [38,39]. ECG and BCG signals were captured with a PEC sensor module (SNGL-I, Zhongke Sengain, Nanjing, China) and two analog front-end chips (AFE1294, Texas Instruments, Dallas, TX, USA), with data transmitted via USART using an STM32F407 microcontroller (STMicroelectronics, Geneva, Switzerland). Customized software displayed and stored all signals on a laptop.

The study received ethics approval from the Chinese PLA General Hospital. The experimental setup is shown in Figure 4a, and an example of CVSs is presented in Figure 4c. DCGs were extracted from the radar’s intermediate frequency (IF) signals using all 2 TX and 4 RX antennas in a MIMO configuration with binary phase modulation (BPM). Signals underwent beamforming, circle fitting, and phase unwrapping to generate DCG signals. For analysis, the three positions with the highest reflection energy were selected for DCG signals, along with a single BCG signal.

All signals were sampled at 250 Hz and upsampled to 1000 Hz using quadratic spline interpolation. The synchronization error between DCGs and BCGs, caused by radar acquisition and processing, was within 40 ms.

#### 4.1.2. Dataset B

Dataset B is an open-source dataset collected and shared by Carlson [40]. The dataset contains several cardiac driving signals, including BCG, ECG, PPG, and arterial blood pressure (ABP) waveforms, from 40 participants. Among them, the eight-channel bed-based BCGs was synchronously acquired using four electromechanical films (EMFi) and four load cells (LC). The EMFis and LCs were placed under the mattress and bedpost, respectively, as shown in Figure 4b. All data were sampled at the same rate of 1000 Hz. In this study, the eight-channel BCG signals in this dataset were used for IBI estimation, and ECGs were used as reference signals. Figure 4d provides an example of the morphological features of BCG signals obtained from EMFi and LC. Note that the data of Subjects #3, #10, #12, #16, #17, and #36 were excluded due to the frequent presence of abnormal heartbeats in the ECGs. Ultimately, 34 individuals were used for evaluation.

The demographic information and IBI distribution for Datasets A and B are presented in Table 1 and Figure 5, respectively.

### 4.2. Metrics

#### 4.2.1. Mean Heart Rate Estimation

To evaluate the performance of the HarSum-HR method used in Section 3.2, we will compare mean heart rates calculated from CVSs and a reference ECG within an 8 s window. The ground truth HR of a window was then computed as:(24)$$HR_{\mathrm{ECG}}=\frac{60}{{\bar{\mathrm{IBI}}}_{\mathrm{ECG}}},$$
where ${\bar{\mathrm{IBI}}}_{\mathrm{ECG}}$ is the mean value of RR intervals of ECG within the window. Christov Segmenter [22], a state-of-the-art method from the BioSPPy Python toolbox [1], was used to label R-peaks in ECG.

The mean absolute error (${\mathrm{MAE}}_{HR}$, bpm) of mean heart rate, the mean relative absolute error (${\mathrm{MRAE}}_{HR}$, %), and the standard deviation of absolute error (${\mathrm{SDAE}}_{HR}$, bpm) were calculated according to Formulas (Equation 25)–(Equation 27).
(25)$${\mathrm{MAE}}_{HR}=\frac{1}{N}\sum_{i=1}^{N}\left|HR_{\mathrm{CVS}}\left(i\right)-HR_{\mathrm{ECG}}\left(i\right)\right|$$
(26)$${\mathrm{MRAE}}_{HR}=\frac{1}{N}\sum_{i=1}^{N}\frac{\left|HR_{\mathrm{CVS}}\left(i\right)-HR_{\mathrm{ECG}}\left(i\right)\right|}{HR_{\mathrm{ECG}}\left(i\right)}\times 100\%$$
(27)$${\mathrm{SDAE}}_{HR}=\sqrt{\frac{1}{N}\sum_{i=1}^{N}{\left(\left|HR_{\mathrm{CVS}}\left(i\right)-HR_{\mathrm{ECG}}\left(i\right)\right|-{\mathrm{MAE}}_{HR}\right)}^{2}},$$
where $HR_{\mathrm{CVS}}\left(i\right)$ represents the mean heart rate of the *i*-th window, and *N* represents the number of windows. Furthermore, the coverage ratio (${\mathrm{Cov}}_{HR}$, %) signifies the proportion of period that satisfies the criteria outlined in Section 3.3 compared to the total recording period.

#### 4.2.2. IBI Estimation

As for evaluating the performance of the proposed HSF-IBI framework, the corresponding MAE, MRAE, and SDAE are calculated.
(28)$${\mathrm{MAE}}_{\mathrm{IBI}}=\frac{1}{N}\sum_{i=0}^{N-1}\left|\mathrm{RR}(k)-\mathrm{JJ}(k)\right|$$
(29)$${\mathrm{MRAE}}_{\mathrm{IBI}}=\frac{1}{N}\sum_{i=1}^{N}\frac{\left|\mathrm{RR}(k)-\mathrm{JJ}(k)\right|}{\mathrm{RR}(k)}\times 100\%$$
(30)$${\mathrm{SDAE}}_{\mathrm{IBI}}=\sqrt{\frac{1}{N}\sum_{i=1}^{N}{\left(\left|\mathrm{RR}(k)-\mathrm{JJ}(k)\right|-{\mathrm{MAE}}_{\mathrm{IBI}}\right)}^{2}}.$$

In addition, the precision (${\mathrm{Prec}}_{\mathrm{IBI}},\%$) of the detected IBI is defined as follows:(31)$${\mathrm{Prec}}_{\mathrm{IBI}}=\frac{\mathrm{correct}}{\mathrm{incorrect}+\mathrm{correct}}\times 100\%.$$

In Equation (Equation 31), the classification of correct and incorrect is determined by whether the absolute error between the J-J interval and the R-R interval falls within 30 ms or not [28,41].

The IBI detection rate (${\mathrm{Det}}_{\mathrm{IBI}},\%$) is defined as the measure of how many heartbeats’ IBIs are accurately detected out of all heartbeats. Here, all heartbeats refer to the total number of heartbeats detected from the ECG, including those removed due to low CVS quality in Section 3.3.
(32)$${\mathrm{Det}}_{\mathrm{IBI}}=\frac{\mathrm{correct}}{\mathrm{all}\;\mathrm{heartbeats}}\times 100\%.$$

### 4.3. Evaluation Scenario

To evaluate the performance improvement of heterogeneous sensors’ fusion in heart rate measurement and IBI estimation, the input signals were adjusted and analyzed for single-channel (SC), single-type sensor multi-channel (MC), and heterogeneous sensor multi-channel (HS) scenarios. In the single-channel case, the cardiac vibration signal closest to the heart from the various channels of the sensors was selected. Furthermore, to assess the role of the OBS module, the results without using the OBS module were tested in the HS scenario (HS-NOBS). In HS-NOBS scenario, during the preprocessing stage of the algorithm (Section 3.1), only the low cut-off frequency of $f_{L}=3$ Hz for the band-pass filter was retained.

## 5. Results and Discussion

### 5.1. Mean HR Estimation: The Performance of HarSum-HR

Table 2 presents the statistical results of short-term mean heart rates obtained using the HarSum-HR algorithm. The displayed results represent the average results for all participants in the datasets. All mean heart rates meet the Association for the Advancement of Medical Instrumentation (AAMI) recommended accuracy standards for heart rate measurement (within ±5 bpm).

In Dataset A, the SC-Radar scenario exhibits the highest ${\mathrm{MAE}}_{HR}$ (1.54 bpm) and ${\mathrm{MRAE}}_{HR}$ (1.75%) as well as the lowest ${\mathrm{Cov}}_{HR}$ (87.23%). This is due to the overall lower quality of the DCG signal obtained from the radar compared to the BCG acquired from the PEC sensor. In Dataset B, there is no significant difference in performance between the two types of sensor.

We can observe that for both Dataset A and Dataset B, the HS scenario significantly improves ${\mathrm{Cov}}_{HR}$ (93.22% for Dataset A and 97.83% for Dataset B) compared to the SC scenario, without noticeably increasing ${\mathrm{MAE}}_{HR}$. This demonstrates the effectiveness of channel fusion in enhancing signal coverage.

In both Datasets A and B, HS does not achieve the optimal ${\mathrm{MAE}}_{HR}$. However, this does not imply a decline in heart rate measurement performance. Since the calculation of mean heart rate uses ECG as the ground truth, there is an inherent systematic error. Specifically, ECG calculates the average heart rate by considering the R-R intervals within a window rather than using a frequency domain method that takes into account the entire window signal. Therefore, this systematic error originates from the individual’s short-term heart rate variability.

As mentioned earlier, introducing the harmonic method to calculate the mean heart rate before computing the IBIs serves the following purposes: it provides a reference quality index for channel selection and fusion and offers a reference range for parameters such as step size when using template matching to calculate the IBIs in subsequent steps.

In some studies, HR is calculated by focusing on specific harmonics of CVSs. For instance, in [42], the HR from the fiber optic sensor is determined by detecting the third harmonic in the frequency domain. Similarly, in [43,44], the HR from the radar DCG signals is calculated by focusing on the second or third harmonic. However, when considering heterogeneous sensors and a greater variety of subject positions and postures, the optimal harmonic order is not fixed. HarSum-HR offers a highly adaptable solution to calculate the mean HR under various signal morphologies. Figure 6 demonstrates the stability of HarSum-HR in calculating average heart rate. When postural changes caused alterations in signal morphology, the wavelet-based method experienced frequency doubling errors, identifying the second harmonic as the heart rate. In contrast, HarSum-HR consistently maintained stable heart rate tracking. In addition, HarSum-HR effectively increases heart rate extraction accuracy by considering the overall energy of harmonics, which is more reliable than focusing solely on the fundamental frequency, which may sometimes shift in the spectrum due to influences such as respiratory harmonics. Additionally, it is noteworthy that this algorithm, with the same parameter settings, can effectively calculate the heart rate from both low-frequency PPG signals and high-frequency phonocardiography (PCG) signals.

### 5.2. IBI Estimation

#### 5.2.1. Effectiveness of Heterogeneous Sensors Fusion

The results for the SC and MC scenarios in Table 3 show that the multi-channel fusion of a single sensor type can improve the accuracy and coverage of IBI estimation, as previously proven by earlier research [16,21].

On the other hand, the results for the MC and HS scenarios in Table 3 indicate that the introduction of heterogeneous sensors can further enhance performance. Compared to the highest results in other scenarios, the detection rate in the HS scenario has increased 6.25% (from 89.15% to 95.40%) and 5.21% (from 92.35% to 97.56%) in Datasets A and B, respectively. The reason for this improvement is that, in addition to the spatial distribution, another crucial factor is the varying sensitivity of different types of sensors to various types of noise, as it is often observed that all channels of one sensor type may be influenced by a specific noise while another sensor type maintains high-quality signals.

In terms of accuracy, the results of heterogeneous sensors generally outperform those of single-type sensors. However, an exception occurs with radar and piezoelectric sensors. This is because the radar’s ${\mathrm{MAE}}_{IBI}$ is generally higher than that of the PEC sensors. When heterogeneous sensors are fused to improve coverage, the introduction of radar slightly reduces the overall accuracy.

Figure 7 presents the Bland–Altman analysis of the errors between ${\mathrm{IBI}}_{ECG}$ and ${\mathrm{IBI}}_{CVS}$ across various scenarios. In all cases, the majority of IBI errors fall within the 95% limits of agreement, suggesting the overall reliability of the algorithm employed in this study. More specifically, in the example from Dataset A (Figure 7a), the SC-PEC scenario achieves the smallest width of the limits of agreement (24.70 ms) and the lowest ${\mathrm{MAE}}_{IBI}$ (5.21 ms), while the SC-Radar has the largest width and ${\mathrm{MAE}}_{IBI}$. This indicates that the nature of the sensor plays a dominant role in terms of accuracy. In the HS scenario, the width of the limits of agreement (27.30 ms) and ${\mathrm{MAE}}_{IBI}$ (5.41 ms) results are slightly worse than the best-case scenario; however, there is an improvement in the ${\mathrm{Det}}_{IBI}$ value (89.09–94.72%). This outcome can be viewed as a result of combining IBI measurements from both sensor types, where the inclusion of some IBIs from the radar slightly reduces accuracy compared to SC-PEC in exchange for a higher detection rate. Similar findings can also be observed in the example from Dataset B (Figure 7b).

#### 5.2.2. Effectiveness of OBS Module

Comparing the HS and HS-NOBS scenario in Table 3, we observe that the ${\mathrm{MAE}}_{IBI}$ increases from 5.45 ms to 12.39 ms on Dataset A and from 4.56 ms to 8.21 ms on Dataset B, highlighting the enhanced accuracy achieved by introducing the OBS module. This improvement can be explained by the heart valve signal [31]. After removing low-frequency components, the higher frequency CVS signals are caused by heart valve closure, and this phenomenon becomes more evident as the distance to the heart decreases. In cases of suboptimal positioning or low signal quality, these high-frequency signals may be less noticeable or absent. Similar to BCG signals in [31], it has also been validated that heart sounds in radar-measured DCGs can be obtained through higher frequency filtering [32,45]. The occurrence of these high-frequency components is closer to the sinus node, making them less affected by changes in vascular status and resulting in lower variability.

On the other hand, it has been demonstrated that when using a variable low-pass cutoff frequency filter for BCG filtering, the R-J interval variability increases as the cutoff frequency decreases [46]. This can be attributed to the random effect of the waveform peak flattening caused by excessive filtering. Similarly, in scenarios in which the OBS module is not used and low-frequency components dominate the signal, the relatively flat peaks lead to a more random detection of peak positions.

In this study, we chose to fix the high-pass cutoff frequency at 40Hz and adjust the low-pass cutoff frequency to select frequency bands. This is because we observed that, in CVSs, low-frequency components (<10 Hz) occupy more energy than high-frequency components (>10 Hz), and a low-quality signal typically does not contain periodic high-frequency components. This means that for the signal of each channel, we need to determine whether it contains the desired high-frequency periodic components. The function of the OBS module is to select filtered signals that satisfy the threshold based on the SQI. To avoid phase distortion, the filtering process should be applied in both forward and reverse directions, as phase distortion can also contribute to an increased standard deviation in R-J intervals.

Figure 8 illustrates an example of how the OBS module functions on a single-channel BCG signal. Figure 8a demonstrates that when a 1–40 Hz band-pass filter is applied, the detected heartbeat positions correspond to the maximum value locations of each heartbeat in the original signal; conversely, when an 8–40 Hz band-pass filter is used, the final detected heartbeat positions correspond to a small peak in the original signal. The respective IBI results are depicted in Figure 8b, and when compared to the ECG’s IBI, the ${\mathrm{MAE}}_{IBI}$ obtained from the BCG signals for the two filtering frequencies are 12.10 ms and 3.00 ms, respectively.

### 5.3. Comparison with Other Works

#### 5.3.1. Comparison on the Same Dataset

First, we compared our work with two state-of-the-art methods for extracting IBI from multichannel CVS. xCLIE [16] uses Bayesian fusion to combine the results of three time-domain estimators across multiple channels, while weight xCLIE [21] employs a Gaussian weighting curve to incorporate previous IBI estimation results and uses Q-values for channel selection. We replicated these methods using the parameters provided in the original papers. The mean and standard deviation of ${\mathrm{MAE}}_{IBI}$ and ${\mathrm{DET}}_{IBI}$ results of all subjects are shown in Figure 9. Weight xCLIE (${\mathrm{MAE}}_{IBI}$: 12.56 ± 4.10 ms/9.15 ± 3.2 ms, ${\mathrm{DET}}_{IBI}$: 93.26 ± 6.21%/90.21 ± 7.28%) performs better than xCLIE (${\mathrm{MAE}}_{IBI}$: 15.36 ± 4.91 ms/11.45 ± 4.20 ms, ${\mathrm{DET}}_{IBI}$: 81.56 ± 6.12%/89.15 ± 6.91%) primarily because it introduces a channel selection step, rather than using all channel information. However, it still uses Bayesian fusion of multiple time-domain estimators for single-channel IBI calculation, which is not optimal. The proposed HSF-IBI method shows significantly better performance (${\mathrm{MAE}}_{IBI}$:5.25 ± 3.51 ms/4.56 ± 2.90 ms, ${\mathrm{DET}}_{IBI}$: 92.40 ± 6.38%/97.56 ± 6.20%) in terms of ${\mathrm{MAE}}_{IBI}$ across both datasets. This is mainly because the proposed framework avoids Bayesian methods that are not based on precise feature point localization but rather estimate the most probable IBI based on probability density, potentially introducing errors from low-quality channel signals and estimators into the results. In terms of coverage, using a 30 ms threshold for correct detection, the proposed method also achieves a relatively high level.

#### 5.3.2. Comparison with Results from Other Studies

Table 4 presents the primary performance results of various works available to date. It is evident from the error metrics for IBI detection (such as MAE, MRAE, SD, Median, etc.) that the results from this work are superior when compared with those of other studies, even using the SC scenario from this paper (with MAE values of 11.65 ms, 5.25 ms, 6.46 ms, and 6.04 ms) and the MC scenario (with MAE values of 10.41 ms, 4.21 ms, and 5.38 ms). This demonstrates that even without utilizing multi-sensor data, the proposed TM+OBS+SQI algorithm for precise IBI extraction shows excellent performance.

While it is not entirely rigorous to demonstrate the superiority of the proposed method using these results, as they are evaluated on different datasets, another objective of this comparative analysis is to provide a common performance reference for various unobtrusive IBI measurement methods in this field.

For reference, a comparative study [48] that employed the same dataset as Dataset B tested three types of algorithms. The best ${\mathrm{MRAE}}_{IBI}$ on EMFis and load cell in [48] was 2.91% and 2.28%, respectively, while the results for the SC scenarios in this paper were 0.85% and 0.75%.

Compared to deep learning-based IBI measurement methods, the proposed method does not require a prior training phase and does not have concerns regarding generalization performance. Unlike morphology-based methods, our approach does not require prior knowledge of heartbeat waveform morphology but relies solely on the quasi-periodicity of the heartbeat signals. Another advantage of our work is the algorithm’s adaptability to signals from various unobtrusive sensors. The selection of all hyperparameters in this paper is grounded in physiological theory rather than the characteristics of signal morphology. Our algorithm has been tested on four types of unobtrusive sensors (Radar, PEC, EMFi, and load cell) and even on PCG and PPG signals, whereas other studies have typically focused on a single sensor type.

Furthermore, to demonstrate the effect of sampling rate on performance, an additional experiment was conducted. In this experiment, the original unprocessed signal, initially at 1000 Hz, was downsampled by integer factors to predetermined frequencies. Then, on this low-sampling-rate signal, we evaluated the proposed method either directly or after restoring it to a 1000 Hz signal using quadratic spline interpolation. The resulting ${\mathrm{MAE}}_{IBI}$ variations with the sampling frequency are shown in Figure 10.

For Dataset A, when no interpolation is used and the sampling rate is reduced to 200 Hz, the ${\mathrm{MAE}}_{IBI}$ increases from 5.25 ms to 7.30 ms, and at a sampling rate of 100 Hz, the ${\mathrm{MAE}}_{IBI}$ is 8.24 ms. When using quadratic spline interpolation, the corresponding ${\mathrm{MAE}}_{IBI}$ values at 200 Hz and 100 Hz are 5.27 ms and 5.47 ms, respectively. For Dataset B, the ${\mathrm{MAE}}_{IBI}$ values are 5.08 ms (200 Hz, NonInterpolated), 5.80 ms (100 Hz, Non- Interpolated), 4.65 ms (200 Hz, Interpolated), and 4.67 ms (100 Hz, Interpolated). The MAE values obtained for these frequency-shifted results are lower than those in related works, indicating that the sampling rate is not the primary reason for the good results. These results also provide some guidance on the optimal sampling rate for unobtrusive heart rate measurement.

In the results after quadratic spline interpolation, the ${\mathrm{MAE}}_{IBI}$ of Dataset A changes only slightly with the downsampling ratio, whereas the ${\mathrm{MAE}}_{IBI}$ of Dataset B exhibits a noticeable variation. This is because the original signals in Dataset B are more affected by power frequency noise interference, which leads to a decrease in signal quality after downsampling. This experiment also demonstrates that, for unobtrusive IBI estimation using CVSs, adopting a 100 Hz sampling rate in conjunction with suitable interpolation effectively preserves the accuracy of the obtained results.

### 5.4. Limitations

Although this work has achieved satisfactory results, there are still some limitations that need to be addressed. Initially, similar to many other studies in this area, the majority of the data within our datasets were obtained from healthy young participants and collected in controlled laboratory settings. Additionally, despite incorporating multiple datasets and analyzing a large number of heartbeat samples, the overall number of participants remains relatively small. In the future, validating the method with a larger and more diverse participant pool will be crucial for improving the robustness of IBI measurement. Furthermore, exploring modifications to mitigate interference from abnormal heartbeats and tailoring the method for patients with heart conditions will be essential.

Additionally, the integration of multiple channels inevitably increases computational demands. However, our algorithm is designed to minimize computational complexity. By focusing the matching process on peak points, we significantly reduce the most time-consuming aspect of calculating the correlation coefficient in traditional template matching (TM) methods. In practical tests on an embedded Cortex-A7 processor operating at 1 GHz, the algorithm’s execution time was only 1.1% of the duration of the processed signal, demonstrating its feasibility for real-time computation.

## 6. Conclusions

Unobtrusive vital sign monitoring offers unparalleled advantages in terms of comfort compared to other monitoring methods, but its robustness must be considered. Combining different types of sensors is a worthwhile optimization approach.

In this study, we explore the use of unobtrusive heterogeneous sensor fusion for the first time to enhance the performance of inter-beat interval (IBI) estimation and introduce a novel algorithmic framework, HSF-IBI (heterogeneous sensor fusion-IBI). This framework incorporates TM (template matching), HarSum-HR (harmonic summation-heart rate), BSAS (basic sequential algorithmic scheme), OBS (optimal band selection), and SQI (signal quality index)-based beat fusion techniques. The proposed framework does not require prior training or knowledge of heartbeat waveform morphology, and it can integrate various types of data from unobtrusive sensors. Our results demonstrate that the fusion of heterogeneous sensors improves the detection rate of IBI monitoring and that the proposed algorithm outperforms Bayesian-based multi-channel fusion methods.

In summary, the HSF-IBI algorithmic framework provides a solution for achieving more accurate and robust IBI measurement and HRV monitoring.

## Figures and Tables

**Figure 1 bioengineering-11-01219-f001:**
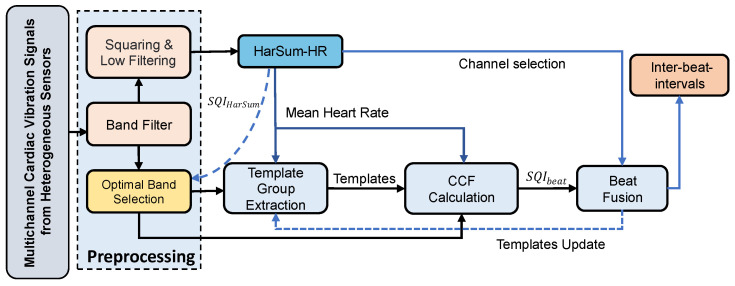
Overview of proposed HSF-IBI method. Multi-channel CVSs are first processed through a set of bandpass filters, followed by another preprocessing step: squaring and low-pass filtering to shift the frequency to a lower position for calculating HarSum spectrums in the HarSum-HR. The HarSum-HR computes the mean heart rate and signal quality of a signal segment (${SQI}_{Harsum}$), and the obtained mean heart rate and signal quality are used as references for selecting the signal channel and optimal band selection (OBS), calculating the template group, determining and selecting the beat-by-beat IBIs. The template group, correlation coefficient function (CCF), and beat-by-beat IBIs are calculated from the signals in the optimal frequency band of the selected satisfactory signal channels. Finally, the ultimate beat-by-beat IBIs are chosen based on ${SQI}_{beat}$.

**Figure 2 bioengineering-11-01219-f002:**
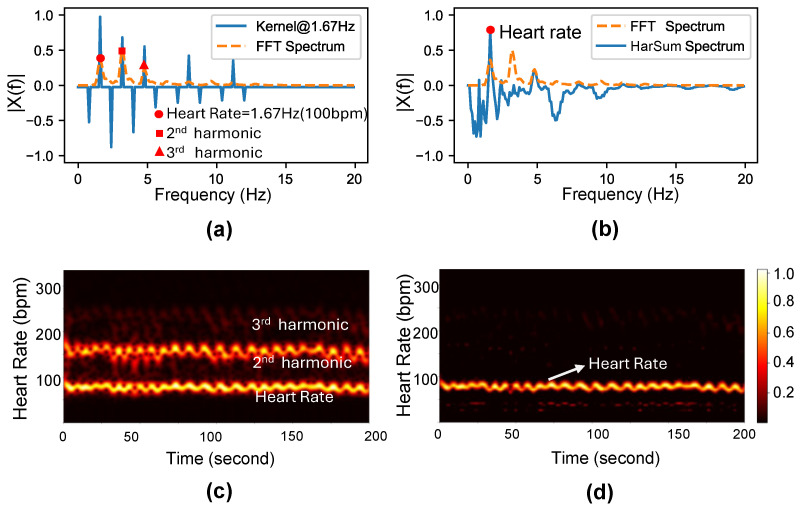
Calculating mean HR from FFT spectrum using HarSum-HR. (**a**) The FFT spectrum of an 8 s BCG signal from a piezoelectric film and the kernel function $K\left(f;f^{\prime }\right)$ at $f^{\prime }=1.67$ Hz. (**b**) The FFT spectrum of the BCG signal and the result after HarSum transformation. (**c**,**d**) show the short-time Fourier transform (STFT) spectrum and the time–frequency spectrum result after HarSum-HR of a 200 s BCG signal from a force sensor.

**Figure 3 bioengineering-11-01219-f003:**
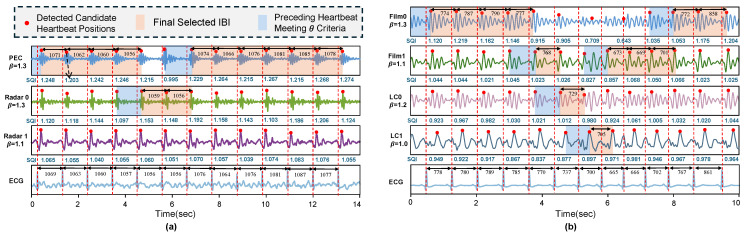
Schematic diagram of inter-beat interval (IBI) fusion across channels based on $SQI_{\mathrm{beat}}$ (signal quality index for each heartbeat). Panels (**a**) and (**b**) display examples from Dataset A and Dataset B, respectively. The last row of signals represents the reference ECG signal, with red dashed lines indicating the reference heartbeats divided by the R-peaks of the ECG. The signals in the rows above are the cardiac vibration signal (CVS) channels selected in Section 3.3. PEC, Radar0, and Radar1 represent CVSs from piezoelectric sensors and two radar channels in Dataset A. Film0, Film1, LC0, and LC1 correspond to electromechanical films and load cells. All IBI values shown in the figure are labeled in milliseconds. The parameter β is the weighting coefficient applied to the $SQI_{\mathrm{beat}}$ of each channel, as defined by the filtering parameters in Equation (Equation 2). The β-weighted $SQI_{\mathrm{beat}}$ values are displayed below each channel, with each SQI value corresponding to a single heartbeat. Overall, the final IBI is selected as the heartbeat with the highest $SQI_{\mathrm{beat}}$ that also meets the threshold requirement (θ=0.75) for the preceding heartbeat.

**Figure 4 bioengineering-11-01219-f004:**
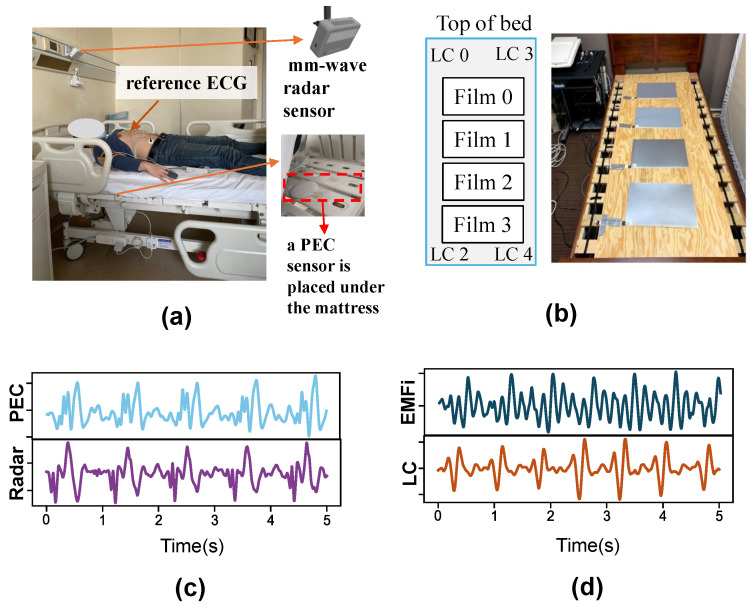
Dataset description. (**a**) Sensor placement diagram for Dataset A: Radar mounted above the subject’s head and PEC sensors positioned beneath the mattress. (**b**) Sensor placement diagram for Dataset B: Four EMFi sensors arrayed on the mattress and four load cells situated under the four bed legs [40]. (**c**) CVSs samples from Dataset A. (**d**) CVS sample from Dataset B.

**Figure 5 bioengineering-11-01219-f005:**
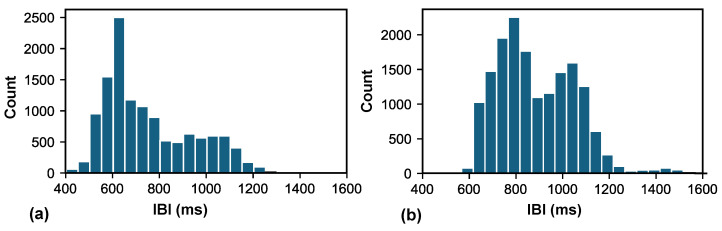
Histogram of IBI distribution. (**a**) Distribution of IBIs for a total of 12,481 heartbeats in Dataset A. (**b**) Distribution of IBIs for a total of 16,375 heartbeats in Dataset B.

**Figure 6 bioengineering-11-01219-f006:**
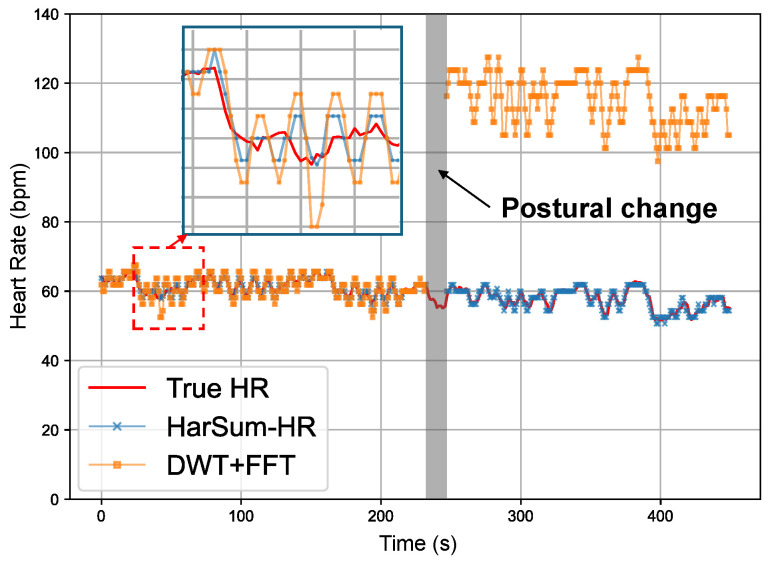
Comparison of the performance of HarSum-HR and the baseline method in calculating mean heart rate. The baseline method employs discrete wavelet transform (DWT) to extract heartbeat components followed by FFT. However, it experienced frequency doubling errors when postural changes altered the signal morphology.

**Figure 7 bioengineering-11-01219-f007:**
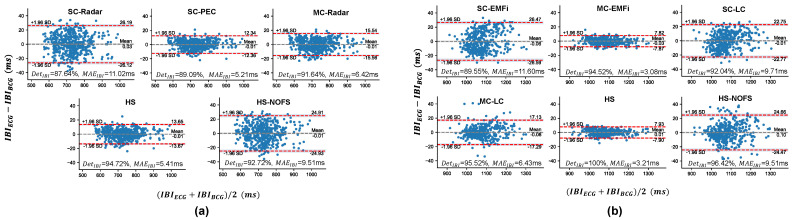
Bland−Altman plots of IBI estimation errors for two exemplary subjects under different scenarios, along with their corresponding ${\mathrm{Det}}_{IBI}$ and ${\mathrm{MAE}}_{IBI}$ values. (**a**) Results for Subject #1 in Dataset A. (**b**) Results for Subject #4 in Dataset B.

**Figure 8 bioengineering-11-01219-f008:**
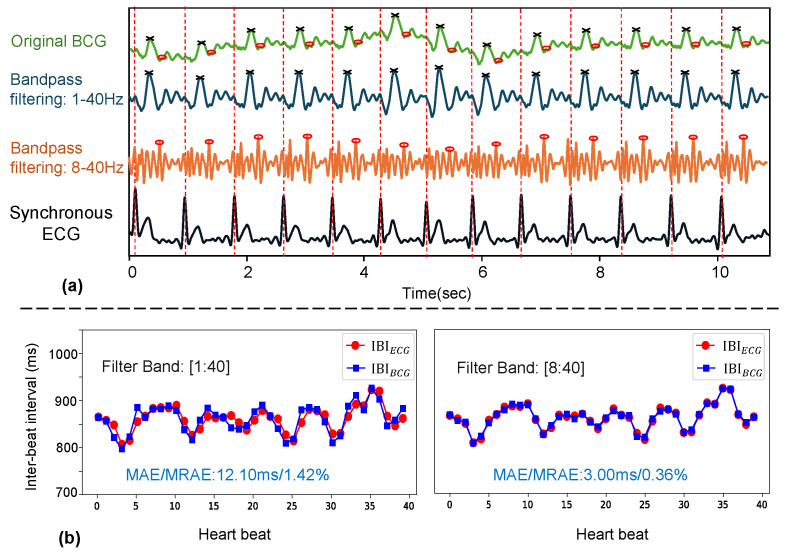
Diagram of the effect of the introduction of OBS on the IBI error. (**a**) The result of a segment of BCG signal processed by bandpass filters with different cutoff frequencies and the detected heartbeat location. The position on the filtered signal is marked with the same mark on the original signal. (**b**) IBI sequence line diagram of signal in (**a**) after processing by different filters.

**Figure 9 bioengineering-11-01219-f009:**
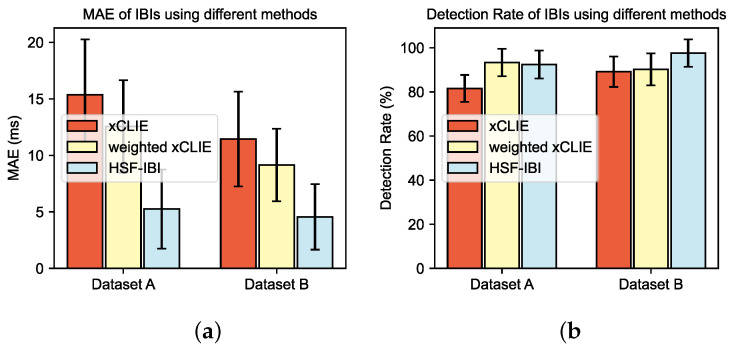
Comparison of the proposed framework with other methods. (**a**) ${\mathrm{MAE}}_{IBI}$ (**b**) ${\mathrm{Det}}_{IBI}$.

**Figure 10 bioengineering-11-01219-f010:**
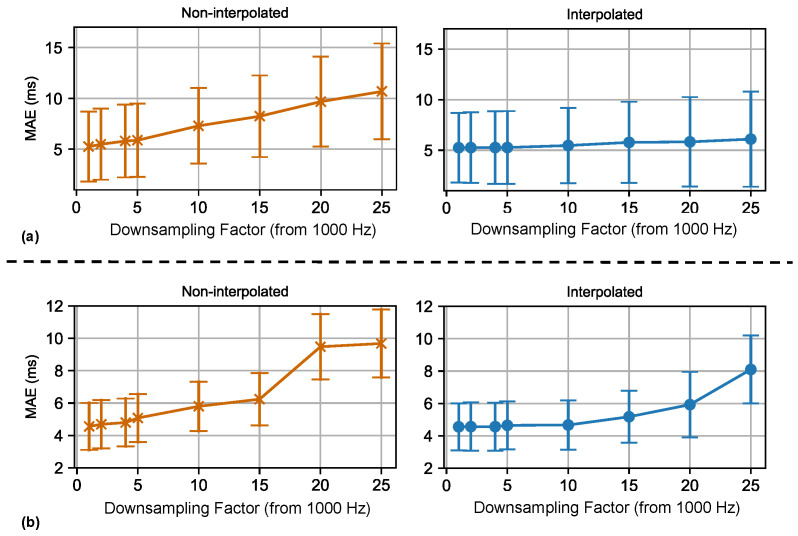
Variation in IBI MAE with downsampling ratio from 1000 Hz. (**a**) Results for Dataset A. (**b**) Results for Dataset B. **Left**: Data downsampled from 1 kHz by integer factors. **Right**: Data downsampled from 1 kHz by integer factors and then interpolated back to 1 kHz using quadratic spline interpolation. Standard deviation between subjects is also shown at each point on the line chart. Downsampling ratios and their corresponding frequencies: 1 (1000 Hz), 2 (500 Hz), 4 (250 Hz), 5 (200 Hz), 10 (100 Hz), 15 (67 Hz), 20 (50 Hz), 25 (40 Hz).

**Table 1 bioengineering-11-01219-t001:** Demographic characteristics of subjects in the datasets.

Characteristics	Dataset A (Radar & PEC)	Dataset B (EMFi & LC)
Number (Male + Female)	19 (15 + 4)	36 (15 + 21)
Age (years)	28 ± 7	34 ± 15
Weight (kg)	68 ± 16	76 ± 18
Height (cm)	172 ± 10	171 ± 11
BMI (kg/m^2^)	23 ± 4.3	26 ± 5.7

**Table 2 bioengineering-11-01219-t002:** Performance of mean heart rate estimation on the two datasets. (Bold highlights the best results, while underlining indicates the second-best).

Dataset	Evaluation Scenario	${\mathrm{MAE}}_{\mathrm{HR}}$ [bpm]	${\mathrm{MRAE}}_{\mathrm{HR}}$ [%]	${\mathrm{SDAE}}_{\mathrm{HR}}$ [bpm]	${\mathrm{Cov}}_{\mathrm{HR}}$ [%]
Dataset A	SC-Radar	1.54	1.75	1.14	87.23
	SC-PEC	**1.34**	**1.52**	0.94	89.15
	MC-Radar	1.36	1.54	1.02	89.40
	HS	1.35	1.53	**0.92**	**93.22**
Dataset B	SC-EMFi	1.57	2.18	1.28	90.90
	SC-LC	**1.49**	**2.04**	1.28	89.41
	MC-EMFi	1.65	2.30	1.39	92.45
	MC-LC	1.55	2.17	1.32	93.56
	HS	1.56	2.23	**1.25**	**97.83**

**Table 3 bioengineering-11-01219-t003:** Performance of IBI estimation on the two datasets. (Bold highlights the best results, while underlining indicates the second-best).

Dataset	Evaluation Scenario	${\mathrm{MAE}}_{\mathrm{IBI}}$ [ms]	${\mathrm{MRAE}}_{\mathrm{IBI}}$ [%]	${\mathrm{SDAE}}_{\mathrm{IBI}}$ [ms]	${\mathrm{Prec}}_{\mathrm{IBI}}$ [%]	${\mathrm{Det}}_{\mathrm{IBI}}$ [%]
Dataset A	SC-Radar	11.65	1.67	6.95	98.47	85.90
	SC-PEC	**5.25**	**0.75**	4.28	**100.00**	89.15
	MC-Radar	10.41	1.45	5.70	98.86	88.38
	HS	5.45	0.78	**4.22**	99.12	**95.40**
	HS-NOBS	12.39	1.77	6.87	94.91	88.47
Dataset B	SC-EMFi	6.46	0.80	3.94	99.32	90.28
	SC-LC	6.04	0.75	4.34	98.53	88.09
	MC-EMFi	**4.21**	**0.53**	**3.40**	**99.89**	92.35
	MC-LC	5.38	0.68	4.55	98.29	91.95
	HS	4.56	0.57	3.57	99.76	**97.56**
	HS-NOBS	8.21	1.03	5.48	95.87	93.79

**Table 4 bioengineering-11-01219-t004:** Performance comparison of various IBI estimation methods.

Paper (Year)	Methods and Data				Performance			
	Channels	Algorithm	Sensor	Sample Rate [Hz]	${\mathrm{Det}}_{\mathrm{IBI}}$ ↑ [%]	${\mathrm{MAE}}_{\mathrm{IBI}}$ ↓ [ms]	${\mathrm{MRAE}}_{\mathrm{IBI}}$ ↓ [%]	Other Indicators of IBI
[4] (2013)	Single	CLIE	EMFi	250	72.69	7.09	0.78	–
[7] (2020)	Single	CLIE+IE	EMFi	100	73.24	12.67	1.22	–
[37] (2019)	Single	TM+HT	PEC	250	99.67	–	4.56	–
[28] (2022)	Single	UNet+BiLSTM	PEC	1000	97.59	10.18	–	–
[47] (2022)	Single	TM	CW Radar	1000		–	–	36 ms (SD)
[3] (2021)	Single	HT+F-PNSD	PEC	125	98.50	9.60	8.20	–
[48] (2021)	Multiple	CSWT	EMFi	1000	93.65	23.63 (2.15 bpm)	2.91	–
[48] (2021)	Multiple	CSWT	Load Cell	1000	95.56	18.51 (1.63 bpm)	2.28	–
[11] (2021)	Multiple	VMD	FMCW Radar	–	96.16	–	–	28 ms (Median)
[13] (2023)	Multiple	TM+DP	FMCW Radar	250	91.22	–	–	12 ms (Median)
[16] (2015)	Multiple	xCLIE	PVDF	50	81.00	–	1.00	–
[21] (2021)	Multiple	Weighted xCLIE	Load Cell	80	97.16	8.72	–	–
[49] (2024)	Multiple	Weighted xCLIE	Earbud	50	95.56	34.5	4.16	–
This paper-DA-SC	Single	TM+OBS	FMCW Radar/PEC	250	85.90/89.15	11.65/5.25	1.67/0.75	
This paper-DB-SC	Single	TM+OBS	EMFi/Load Cell	1000	90.28/88.09	6.46/6.04	0.80/0.75	
This paper-DA-MC	Multiple	TM+OBS+SQI	FMCW Radar	250	88.38	10.41	1.45	8.48 ms (Median)
This paper-DB-MC	Multiple	TM+OBS+SQI	EMFi/Load Cell	1000	92.35/91.95	4.21/5.38	0.53/0.68	–
This paper-DA-HS	Multiple	TM+OBS+SQI	FMCW Radar + PEC	250	92.40	5.25	0.75	6.22 ms (SD) 6.07 ms (RMSE) 4.56 ms (Median)
This paper-DB-HS	Multiple	TM+OBS+SQI	EMFi + Load Cell	1000	97.56	4.56	0.57	5.10 ms (SD) 4.70 ms (RMSE) 4.25 ms (Median)

Abbreviations in the table: CLIE (Continuous Local Interval Estimation), IE (Iterative Estimation), TM (Template Matching), HT (Hilbert Transform), BiLSTM (Bidirectional Long Short-Term Memory), CSWT (Continuous Spline Wavelet Transform), F-PNSD (Fuzzy Positive and Negative Slope Discrimination), VMD (Variational Mode Decomposition), DP (Dynamic Programming), DA/DB(Dataset-A/Dataset-B).

## Data Availability

Dataset A in this study are available from the corresponding author and the first author upon reasonable request. Dataset B is available at IEEE DataPort via the following link: https://doi.org/10.21227/77hc-py84.

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
