# Peer review of "HSF-IBI: A Universal Framework for Extracting Inter-Beat Interval from Heterogeneous Unobtrusive Sensors"

_bioengineering, 2024, doi:10.3390/bioengineering11121219_

Round 1

Reviewer 1 Report

Comments and Suggestions for Authors

In this study, a novel framework, HSF-IBI (Heterogeneous Sensor Fusion Inter-Beat Interval) was proposed, with higher performance IBI measurement on heterogeneous unobtrusive sensors collected CVSs. A novel multi-channel algorithm framework for extracting IBIs from CVSs is designed.

Datasets, including a self-collected dataset and an open-source dataset were analyzed with algorithm.

1) In my opinion, the paper is more verbose in section 2 and especially in section 3. They must be shorter.

2) Experimental results method is described and presented in exhaustive manner but data obtained only from healthy young subjects and gathered in controlled laboratory settings. Can authors present some data for patients with heart problems? 

3) The authors can give an evaluation how change  the algorithm when treating patients with heart problems.

Reviewer 2 Report

Comments and Suggestions for Authors

I will share here some of my comments:

Line 24: The author may want to add a reference here.

Line 28: The test seems to be very enthusiastic with the unobstructive methods. Do these methods have disadvantages from the measurement quality point of view. A clarification would be helpful.

Line 38: The author is not including in his discussion the optical methods integrated in wristband or in other wearable devices for the measurement of the heartbeat. To better situate the presented work, and to define the meaning of unobstructive probably better would be helpful.

Line 46: is it cardia rhythm extraction or signal transmission?

Line 120: second word "an" to be corrected to "a".

Line 154: A definition of coverage would be helpful.

Line 188-189: why the abbreviation OBS. What is the B standing for?

Line 224: The author may want to check the Figure numbering!

Formula 5: The variable dk is not defined in the text.

Figure 3: The figure caption is not describing the graphs with sufficient information, with the meaning of each parameter. In principle, a figure caption should be self-sufficient, this means that there is no need to read the text to understand it!

Paragraph 3.7

There is an incoherence in the variable used: BPFHURSUM is not define. BPFHARSUM is defined. Is it the same? The author may want to verify the description of the filters. Does it correspond to a bandpass filter? Why is the cutoff frequency of the low-pass filter lower than the high-pass filter, especially for a Butterworth filter?

Reviewer 3 Report

Comments and Suggestions for Authors

This manuscript reports a methodology for deriving the interbeat interval from a fusion of data from multiple sensors and applying a harmonic summation. The authors have compared their approach to others reported previously with regards to critical performance parameters and have evaluated their methodology over both a dataset specific to this paper and a publicly available dataset for further comparison. The paper is generally clearly written, and data are well summarised by the included tables and figures. The approach taken will likely improve our understanding of interpretation and quantification from non-contact sensors applied to heart rate monitoring. There are however some minor points that I think the authors should address and these are detailed below.

1.      Page 4 lines 138-139 “Moreover, the data . . . variability of morphology.” I think this could be worded better. I am assuming that you mean that the need for training of the deep learning models makes them dependent upon the quality and depth of the training data. Consider rewording to better clarify your meaning here.

2.      Page 5 line 215 “. . . FFT . . .” I cannot find any definition of this term earlier in the manuscript. Please ensure that all abbreviations are defined at first use in the manuscript.

3.      Equations. At several points in the explanation and presentation of the equations you have let typographical errors slip through that look sloppy and may even cause confusion, e.g. writing ‘Hursum’ instead of ‘HarSum’, writing ‘channal’ instead of ‘channel’. Please correct this.

4.      Figure 3. With the exception of ECG, the labelling used for each of the plots is not clear, i.e. why have they been labelled this way and to what do all the label nomenclatures refer? This labelling should be clarified to ensure that the figure is comprehensible independently of the main text.

5.      Figure 4B. This experimental set-up appears to be part of the Carlson dataset and not something that has been done was part of new research in this paper. As such, if you are reusing images and data from an earlier publication then a proper citation is needed here.

6.      Tables 2 and 3. Clarification should be given in the caption of the meaning of the bold font and underlining in the table.

7.      Figure 10 caption. Change ‘ratio form 1000Hz’ to ‘ratio from 1000 Hz’.

8.      Section 5.4 Do you think that the relatively small sample sizes in the two datasets may be imposing some limitations on how far the data can be interpreted? I think there needs to be a balance between the total number of patients and the total number of heart beats analysed. My observation for these datasets is that whilst you likely have used a reasonable number of heartbeat measurements per patient, your total numbers of patients is a bit small. It would be appropriate to include some succinct commentary on this in this section of the manuscript.
